# Dying Experts: Efficient Algorithms
# with Optimal Regret Bounds

**Hamid Shayestehmanesh**[*]
Department of Computer Science
University of Victoria

**Sajjad Azami**[*]
Department of Computer Science
University of Victoria

**Nishant A. Mehta**
Department of Computer Science
University of Victoria
{hamidshayestehmanesh, sajjadazami, nmehta}@uvic.ca

## Abstract

We study a variant of decision-theoretic online learning in which the set of experts that are available to Learner can shrink over time. This is a restricted version of the well-studied sleeping experts problem, itself a generalization of the fundamental game of prediction with expert advice. Similar to many works in this direction, our benchmark is the ranking regret. Various results suggest that achieving optimal regret in the fully adversarial sleeping experts problem is computationally hard. This motivates our relaxation where any expert that goes to sleep will never again wake up. We call this setting "dying experts" and study it in two different cases: the case where the learner knows the order in which the experts will die and the case where the learner does not. In both cases, we provide matching upper and lower bounds on the ranking regret in the fully adversarial setting. Furthermore, we present new, computationally efficient algorithms that obtain our optimal upper bounds.

## 1 Introduction

Decision-theoretic online learning (DTOL) [13, 20, 21, 6] is a sequential game between a learning agent (hereafter called *Learner*) and Nature. In each round, Learner plays a probability distribution over a fixed set of experts and suffers loss accordingly. However, in wide range of applications, this "fixed" set of actions shrinks as the game goes on. One way this can happen is because experts either get disqualified or expire over time; a key scenario of contemporary relevance is in contexts where experts that discriminate are prohibited from being used due to existing (or emerging) anti-discrimination laws. Two prime examples are college admissions and deciding whether incarcerated individuals should be granted parole; here the agent may rely on predictions from a set of experts in order to make decisions, and naturally experts detected to be discriminating against certain groups should not be played anymore. However, the standard DTOL setting does not directly adapt to this case, i.e., for a given round it does not make sense nor may it even be possible to compare Learner's performance to an expert or action that is no longer available.

Motivated by cases where the set of experts can change, a reasonable benchmark is the *ranking regret* [12, 9], for which Learner competes with the best ordering of the actions (see (1) in Section 2 for a formal definition). The situation where the set of available experts can change in each round is known as the *sleeping experts* setting, and unfortunately, it appears to be computationally hard to

---

[*]equal contribution

obtain a no-regret algorithm in the case of adversarial payoffs (losses in our setting) and adversarial availability of experts [10]. This motivates the question of whether the optimal regret bounds can be achieved efficiently for the case where the set of experts can only shrink, which we will refer to as the "dying experts" setting. Applying the results of [12] to the dying experts problem only gives $\mathcal{O}(\sqrt{TK \log K})$ regret, for $K$ experts and $T$ rounds, and their strategy is computationally inefficient.

In more detail, the strategy in [12] is to define a permutation expert (our terminology) that is identified by an ordering of experts, where a permutation expert's strategy is to play the first awake expert in the ordering. They then run Hedge [6] on the set of all possible permutation experts over $K$ experts. Although this strategy competes with the best ordering, the per-round computation of running Hedge on $K!$ experts is $\mathcal{O}(K^K)$ if naïvely implemented, and the results of [10] suggest that no efficient algorithm — one that uses computation $\mathrm{poly}(K)$ per round — can obtain regret that simultaneously is $o(T)$ and $\mathrm{poly}(K)$. However, in the dying experts setting, we show that many of these $K!$ orderings are redundant and only $\mathcal{O}(2^K)$ of them are "effective". The notion of effective experts (formally defined in Section 3) is used to refer to a minimal set of orderings such that each ordering in the set will behave uniquely in hindsight. The behavior of an ordering is defined as how it uses the initial experts in its predictions over $T$ rounds. Interestingly, it turns out that this structure also allows for an efficient implementation of Hedge which, as we show, obtains optimal regret in the dying experts setting. The key idea that enables an efficient implementation is as follows. Our algorithms group orderings with identical behavior into one group, where there can be at most $K$ groups at each round. When an expert dies, the orderings in one of the groups are forced to predict differently and therefore have to redistribute to the other groups. This splitting and rejoining behavior occurs in a fixed pattern which enables us to efficiently keep track of the weight associated with each group.

In certain scenarios, Learner might be aware of the order in which the experts will become unavailable. For example, in online advertising, an ad broker has contracts with their providers and these contracts may expire in an order known to Learner. Therefore, we will study the problem in two different settings: when Learner is aware of this order and when it is not.

**Contributions.**  Our first main result is an upper bound on the number of effective experts (Theorem 3.1); this result will be used for our regret upper bound in the known order case. Also, in preparation for our lower bound results, we prove a fully non-asymptotic lower bound on the minimax regret for DTOL (Theorem 4.1). Our main lower bounds contributions are minimax lower bounds for both the unknown and known order of dying cases (Theorems 4.2 and 4.4). In addition, we provide strategies to achieve optimal upper bounds for unknown and known order of dying (Theorems 4.3 and 4.5 respectively), along with efficient algorithms for each case. This is in particular interesting since, in the framework of sleeping experts, the results of [10] suggest that no-regret learning is computationally hard, but we show that it is efficiently achievable in the restricted problem. Finally, in Section 5.3, we show how to generalize our algorithms to other algorithms with adaptive learning rates, either adapting to unknown $T$ or achieving far greater forms of adaptivity like in AdaHedge and FlipFlop [5].

All formal proofs not found in the main text can be found in the appendix.

## 2   Background and related work

The DTOL setting [6] is a variant of prediction with expert advice [13, 20, 21] in which Learner receives an example $x_t$ in round $t$ and plays a probability distribution $\boldsymbol{p}_t$ over $K$ actions. Nature then reveals a loss vector $\boldsymbol{\ell}_t$ that indicates the loss for each expert. Finally, Learner suffers a loss $\hat{\ell}_t := \boldsymbol{p}_t \cdot \boldsymbol{\ell}_t = \sum_{i=1}^{K} p_{i,t} \ell_{i,t}$.

In the dying experts problem, we assume that the set of experts can only shrink. More formally, for the set of experts $E = \{e_1, e_2, \dots e_K\}$, at each round $t$, Nature chooses a non-empty set of experts $E_a^t$ to be available such that $E_a^{t+1} \subseteq E_a^t$ for all $t \in \{1, \dots, T-1\}$. In other words, in some rounds Nature sets some experts to sleep, and they will never be available again. Similar to [12, 11, 10], we adopt the ranking regret as our notion of regret. Before proceeding to the definition of ranking regret, let us define $\pi$ to be an ordering over the set of initial experts $E$. We use the notion of orderings and permutation experts interchangeably. Learner can now predict using $\pi \in \Pi$, where $\Pi$ is the set of all the orderings. Also, denote by $\sigma^t(\pi)$ the first alive expert of ordering $\pi$ in round $t$; expert $\sigma^t(\pi)$ is the action that will be played by $\pi$. The cumulative loss of an ordering $\pi$ with respect to the available

experts $E_a^t$ is the sum of the losses of $\sigma^t(\pi)$ at each round. We can now define the ranking regret:

$$R_\Pi(1,T) = \sum_{t=1}^T \hat{\ell}_t - \min_{\pi \in \Pi} \sum_{t=1}^T \ell_{\sigma^t(\pi),t} \ . \tag{1}$$

Since we will use the notion of classical regret in our proofs, we also provide its formal definition:

$$R_E(1,T) = \sum_{t=1}^T \hat{\ell}_t - \min_{i \in [K]} \sum_{t=1}^T \ell_{i,t} \ . \tag{2}$$

We use the convention that the subscript of a regret notion $R$ represents the set of experts against which we compare Learner's performance. Also, the argument in parentheses represents the set of rounds in the game. For example, $R_\Pi(1,T)$ represents the regret over rounds 1 to $T$ with the comparator set being all permutation experts $\Pi$. Also, we assume that $\ell_{i,t} \in [0,1]$ for all $i \in [K], t \in [T]$.

Similar to the definition of $E_a^t$, let $E_d^t := E \setminus E_a^t$ be the set of dead experts at the start of round $t$. We refer to a round as a "night" if any expert becomes unavailable on the next round. A "day" is defined as a continuous subset of rounds where the subset starts with a round after a night and ends with a night. As an example, if any expert become unavailable at the beginning of round $t$, we refer to round $t-1$ as a night (and we say the expert dies on that night) and the set of rounds $\{t, t+1 \ldots, t'\}$ as a day, where $t'$ is the next night. We denote by $m$ the number of nights throughout a game of $T$ rounds.

**Related work.** The papers [7] and [1] initiated the line of work on the sleeping experts setting. These works were followed by [2], which considered a different notion of regret and a variety of different assumptions. In [7], the comparator set is the set of all probability vectors over $K$ experts, while we compare Learner's performance to the performance of the best ordering. In particular, the problem considered in [7] aims to compare Learner's performance to the best mixture of actions, which also includes our comparator set (orderings). However, in order to recover an ordering as we define, one needs to assign very small probabilities to all experts except for one (the first alive action), which makes the bound in [7] trivial. As already mentioned, we assume the set $E_a^t$ is chosen adversarially (subject to the restrictions of the dying setting), while in [11] and [15] the focus is on the (full) sleeping experts setting with adversarial losses but *stochastic* generation of $E_a^t$.

For the case of adversarial selection of available actions (which is more relevant to the present paper), [12] studies the problem in the cases of stochastic and adversarial rewards with both full information and bandit feedback. Among the four settings, the adversarial full-information setting is most related to our work. They prove a lower bound of $\Omega(\sqrt{TK \log K})$ in this case and a matching upper bound by creating $K!$ experts and running Hedge on them, which, as mentioned before, requires computation of order $\mathcal{O}(K^K)$ per round. They prove an upper bound of $\mathcal{O}(K\sqrt{T \log K})$ which is optimal within a log factor for the bandit setting using a similar transformation of experts. A similar framework in the bandits setting introduced in [4] is called "mortal bandits"; we do not discuss this work further as the results are not applicable to our case, given that they do not consider adversarial rewards. There is also another line of work which considers the contrary direction of the dying experts game. The setting is usually referred to as "branching" experts, in which the set of experts can only expand. In particular, part of the inspiration for our algorithms came from [8, 14].

The hardness of the sleeping experts setting is well-studied [10, 9, 12]. First, [12] showed for a restricted class of algorithms that there is no efficient no-regret algorithm for sleeping experts setting unless $RP = NP$. Following this, [10] proved that the existence of a no-regret efficient algorithm for the sleeping experts setting implies the existence of an efficient algorithm for the problem of PAC learning DNFs, a long-standing open problem. For the similar but more general case of online sleeping combinatorial optimization (OSCO) problems, [9] showed that an efficient and optimal algorithm for "per-action" regret in OSCO problems implies the existence of an efficient algorithm for PAC learning DNFs. Per-action regret is another natural benchmark for partial availability of actions for which the regret with respect to an action is only considered in rounds in which that action was available.

## 3  Number of effective experts in dying experts setting

In this section, we consider the number of effective permutation experts among the set of all possible orderings of initial experts. The idea behind this is that, given the structure in dying experts, not

all the orderings will behave uniquely in hindsight. Formally, the *behavior* of $\pi$ is a sequence of predictions $(\sigma^1(\pi), \sigma^2(\pi), \ldots, \sigma^T(\pi))$. This means that the behaviors of two permutation experts $\pi$ and $\pi'$ are the same if they use the same initial experts in *every* round. We define the set of effective orderings $\mathcal{E} \subseteq \Pi$ to be a set such that, for each unique behavior of orderings, there only exists one ordering in $\mathcal{E}$.

To clarify the definition of unique behavior, suppose initial expert $e_1$ is always awake. Then two orderings $\pi_1 = (e_1, e_2, \ldots)$ and $\pi_2 = (e_1, e_3, \ldots)$ will behave the same over all the rounds, making one of them redundant. Let us clarify that behavior is not defined based on losses, e.g., if $\pi_1 = (e_i, \ldots)$ and $\pi_2 = (e_j, \ldots)$ where $i \neq j$ both suffer identical losses over all the rounds (i.e. their performances are equal) while using different original experts, then they are not considered redundant and hence both of them are said to be *effective*.

Let $d_i$ be the number of experts dying on the $i^{\text{th}}$ night. Denote by $A$ the number of experts that will always be awake, so that $A = K - \sum_{i=1}^m d_i$. We are now ready to find the cardinality of set $\mathcal{E}$.

**Theorem 3.1.** *In the dying experts setting, for $K$ initial experts and $m$ nights, the number of effective orderings in $\Pi$ is $f(\{d_1, d_2, \ldots d_m\}, A) = A \cdot \prod_{s=1}^m (d_s + 1)$.*

In the special case where no expert dies ($m = 0$), we use the convention that the (empty) product evaluates to 1 and hence $f(\{\}, A) = A$. We mainly care about $|\mathcal{E}|$ as we use it to derive our upper bounds; hence, we should find the maximum value of $f$. We can consider the maximum value of $f$ in three regimes.

1. In the case of a fixed number of nights $m$ and fixed $A$, the function $f$ is maximized by equally spreading the dying experts across the nights. As the number of dying experts might not be divisible by the number of nights, some of the nights will get one more expert than the others. Formally, the maximum value is $(\lceil \frac{D}{m} \rceil^{D \bmod m} \cdot \lfloor \frac{D}{m} \rfloor^{m - (D \bmod m)} \cdot A)$, where $D = K - A + m$ and $K - A \leq m$.

2. In the case of a fixed number of dying experts (fixed $A$), the maximum value of $f$ is $(2^{K-A} \cdot A)$ which occurs when one expert dies on each night. The following is a brief explanation on how to get this result. Denote by $B = (d_1, d_2, \ldots, d_b)$ a sequence of numbers of dying experts where more than one expert dies on some night and $B$ maximizes $f$ (for fixed $A$), so that $F = f(\{d_1, d_2, \ldots, d_b\}, A)$. Without loss of generality, assume that $d_1 > 1$. Split the first night into $d_1$ days where one expert dies at the end of each day (and consequently each of those days becomes a night). Now $F' = f(\{1, 1, \ldots, 1, d_2, \ldots, d_b\}, A)$ where 1 is repeated $d_1$ times. If $d_1 > 1$ then $F' = F \cdot 2^{d_1}/(d_1 + 1) > F$. We see that by splitting the nights we can achieve a larger effective set.

3. In the case of a fixed number of nights $m$, similar to the previous cases, the maximum value is obtained when each night has equal impact on the value of $f$, i.e., when $A = d_1 + 1 = d_2 + 1 = \cdots = d_m + 1$; however, it might not be possible to distribute the experts in a way to get this, in which case we should make the allocation $\{A, d_1 + 1, d_2 + 1, \ldots, d_m + 1\}$ as uniform as possible.

By looking at cases 2 and 3, we see that by increasing $m$ and the number of dying experts, we can increase $f$; thus, the maximum value of $f$ with no restriction is $2^{K-1}$ and is achieved when $m = K - 1$ and $A = 1$.

## 4   Regret bounds for known and unknown order of dying

In this section, we provide lower and upper bounds for the cases of unknown and known order of dying. In order to prove the lower bounds, we need a non-asymptotic minimax lower bound for the DTOL framework, i.e., one which holds for a finite number of experts $K$ and finite $T$. During the preparation of the final version of this work, we were made aware of a result of Orabona and Pál (see Theorem 8 of [16]) that does give such a bound. However, for completeness, we present a different fully non-asymptotic result that we independently developed; this result is stated in a simpler form and admits a short proof (though we admit that it builds upon heavy machinery). We then will prove matching upper bounds for both cases of unknown and known order of dying.

## 4.1 Fully non-asymptotic minimax lower bound for DTOL

We analyze lower bounds on the minimax regret in the DTOL game with $K$ experts and $T$ rounds. We assume that all losses are in the interval $[0, 1]$. Let $\Delta_K := \Delta([K])$ denote the simplex over $K$ outcomes. The minimax regret is defined as

$$\inf_{\boldsymbol{p}_1 \in \Delta_K} \sup_{\boldsymbol{\ell}_1 \in [0,1]^K} \cdots \inf_{\boldsymbol{p}_T \in \Delta_K} \sup_{\boldsymbol{\ell}_T \in [0,1]^K} \left\{ \sum_{t=1}^{T} \boldsymbol{p}_t \cdot \boldsymbol{\ell}_t - \min_{j \in [K]} \sum_{t=1}^{T} \ell_{j,t} \right\}. \tag{3}$$

**Theorem 4.1.** *For a universal constant L, the minimax regret* (3) *is lower bounded by*

$$\frac{1}{L} \min \left\{ \sqrt{(T/2) \log K}, T \right\}.$$

The proof (in the appendix) begins similarly to the proof of the often-cited Theorem 3.7 of [3], but it departs at the stage of lower bounding the Rademacher sum; we accomplish this lower bound by invoking Talagrand's Sudakov minoration for Bernoulli processes [17, 18].

## 4.2 Unknown order of dying

For the case where Learner is not aware of the order in which the experts die, we prove a lower bound of $\Omega(\sqrt{mT \log K})$. Given that we have $E_a^{t+1} \subseteq E_a^t$, the construction for the lower bound of [12] cannot be applied to our case. In other words, our adversary is much weaker than the one in [12], but, surprisingly, we show that the previous lower bound still holds (by setting $m = K$) even with the weaker adversary. We then analyze a simple strategy to achieve a matching upper bound.

In this section, we further assume that $\sqrt{T/2 \log K} < T$ for every $T$ and $K$ so that there is hope to achieve regret that is sublinear with respect to $T$. We now present our lower bound on the regret for the case of unknown order of dying.

**Theorem 4.2.** *When the order of dying is unknown, the minimax regret is* $\Omega(\sqrt{mT \log K})$.

*Proof Sketch.* We construct a scenario where each day is a game decoupled from the previous ones. This means that the algorithm will be forced to have no prior information about the experts at the beginning of each day. First, partition the $T$ rounds into $m + 1$ days of equal length. The days are split into two halves. On the first half, each expert suffers loss drawn i.i.d. from a Bernoulli distribution with $p = 1/2$. At the end of the first half of the day, we choose the expert with the lowest cumulative loss until that round, and that expert will suffer no loss on the second half. For any other expert $e_i$, we use the loss $\ell_{i,t}^{(1)}$ of $e_i$ on the $t^{\text{th}}$ round of the first half to define the loss $\ell_{i,t}^{(2)}$ of $e_i$ on the $t^{\text{th}}$ round of the second half; specifically, we choose the setting $\ell_{i,t}^{(2)} := 1 - \ell_{i,t}^{(1)}$. We show that the ranking regret of the set of orderings over $T$ rounds is obtained by summing the classical regrets of each day over the set of days. $\square$

A natural strategy in the case of unknown dying order is to run Hedge over the set of initial experts $E$ and, after each night, reset the algorithm. We will refer to this strategy as "Resetting-Hedge". Theorem 4.3 gives an upper bound on regret of Resetting-Hedge.

**Theorem 4.3.** *Resetting-Hedge enjoys a regret of* $R_\Pi(1, T) = \mathcal{O}(\sqrt{mT \log K})$.

*Proof.* Let $\tau_s$ be the set of round indices of day $s$; hence, we have $\sum_{s=1}^{m+1} |\tau_s| = T$. The overall ranking regret can be upper bounded by the sum of classical regrets for every interval. Hence, the analysis is as follows:

$$R_\Pi(1, T) \leq \sum_{s=1}^{m+1} \sqrt{|\tau_s| \log(K - s)} \leq \sqrt{\log K} \sum_{s=1}^{m+1} \sqrt{|\tau_s|} \leq \sqrt{(m+1)T \log K}; \tag{4}$$

the last inequality is essentially from the Cauchy-Schwarz inequality (see Lemma B.2). $\square$

Although the basic Resetting-Hedge strategy adapts to $m$, it has many downsides. For example, resetting can be wasteful in practice. Another natural strategy, simply running Hedge on the set of

all $K!$ permutation experts, is non-adaptive (obtaining regret $\mathcal{O}(\sqrt{TK \log K})$ and computationally inefficient if implemented naïvely). However, as we show in Section 5.1, this algorithm can be implemented efficiently (with runtime linear in $K$ rather than $K!$) and also, as we show in Section 5.3, by running Hedge on top of several copies of Hedge (one per specially chosen learning rate), we can obtain a guarantee that is far better than Theorem 4.3. Moreover, our efficient implementation of Hedge can be extended to adaptive algorithms like AdaHedge and FlipFlop [5].

### 4.3 Known order of dying

A natural question is whether Learner can leverage information about the order of experts that are going to die to achieve a better regret. We show that the answer is positive: the bound can be improved by a logarithmic factor. We also give a matching lower bound for this case (so both bounds are tight).

Similar to the unknown setting, we provide a construction to prove a lower bound on the ranking regret in this case. We still assume that $\sqrt{T/2 \log K} < T$.

**Theorem 4.4.** *When Learner knows the order of dying, the minimax regret is* $\Omega(\sqrt{mT})$.

*Proof Sketch.* Our construction involves first partitioning all the rounds to $m/2$ days of equal length. On day $s$, all experts will suffer loss 1 on all the rounds except for experts $e_{2s-1}$ and $e_{2s}$, who will suffer losses drawn i.i.d. from a Bernoulli distribution with success probability $p = 1/2$. Experts $e_{2s-1}$ and $e_{2s}$ will die at the end of day $s$, and therefore, each "day game" effectively has 2 experts; our lower bound holds even when Learner knows this fact. Furthermore, Learner will be aware that these two experts ($e_{2s-1}$ and $e_{2s}$) will die at the end of day $s$. Similar to the proof of Theorem 4.2, the minimax regret is lower bounded by the sum of the minimax regrets over each day game. □

Although the proof is relatively simple, it is at least a little surprising that knowing such rich information as the order of dying only improves the regret by a logarithmic factor.

To achieve an optimal upper bound, using the results of Theorem 3.1, the strategy is to create $2^m(K - m)$ experts (those that are effective) and run Hedge on this set.

**Theorem 4.5.** *For the case of known order of dying, the strategy as described above achieves a regret of* $\mathcal{O}\big(\sqrt{T(m + \log K)}\big)$.

*Proof.* Hedge has regret of $\mathcal{O}(\sqrt{T \log K})$ for $K$ number of experts. Therefore, running Hedge on $2^m(K - m)$ experts yields the desired bound. □

Though the order of computation in the above strategy is better than $\mathcal{O}(K^K)$, it is still exponential in $K$. In the next section, we introduce algorithms that simulate these strategies but in a computationally efficient way.

## 5 Efficient algorithms for dying experts

The results of [10] imply computational hardness of achieving no-regret algorithms in sleeping experts; yet, we are able to provide efficient algorithms for dying experts in the cases of unknown and known order of dying. For the sake of simplicity, we initially assume that only one expert dies each night. Later, in Section 5.3, we show how to extend the algorithms for the general case where multiple experts can die each night. We then show how to extend these algorithms to adaptive algorithms such as AdaHedge [5]. The algorithms for both cases are given in Algorithms 1 and 2.

### 5.1 Unknown order of dying

We now show how to efficiently implement Hedge over the set of all the orderings. Even though Resetting-Hedge is already efficient and achieves optimal regret, it has its own disadvantages. The issue arises when one needs to extend Resetting-Hedge to adaptive algorithms. This is particularly important in real-world scenarios, where Learner wants to adapt to the environment (such as stochastic or adversarial losses). We show that Algorithm 1, Hedge-Perm-Unknown (HPU), can be adapted to AdaHedge [19] and, therefore, we can simulate FlipFlop [5]. Next, we give the main idea on how the

| **Algorithm 1:** Hedge-Perm-Unknown (HPU) | **Algorithm 2:** Hedge-Perm-Known (HPK) |
|---|---|
| $\forall i \in [K]\ c_{i,1} := 1,\ h_{i,1} := (K-1)!$ | $\forall i \in [K]\ c_{i,1} := 1,\ h_{i,1} := \lceil 2^{K-i-1} \rceil$ |
| $E_a := \{e_1, e_2, \ldots e_K\}$ | $E_a := \{e_1, e_2, \ldots e_K\}$ |
| **for** $t = 1, 2, \ldots, T$ **do** | **for** $t = 1, 2, \ldots, T$ **do** |
| $\quad$ play $\mathbf{p}_t = \left[ \mathbf{1}\left[e_i \in E_a\right] \cdot \frac{h_{i,t} \cdot c_{i,t}}{\sum_{j=1}^{k} h_{j,t} \cdot c_{j,t}} \right]_{i \in [K]}$ | $\quad$ play $\mathbf{p}_t = \left[ \mathbf{1}\left[e_i \in E_a\right] \cdot \frac{h_{i,t} \cdot c_{i,t}}{\sum_{j=1}^{k} h_{j,t} \cdot c_{j,t}} \right]_{i \in [K]}$ |
| $\quad$ receive $(\ell_{1,t}, \ldots, \ell_{K,t})$ | $\quad$ receive $(\ell_{1,t}, \ldots, \ell_{K,t})$ |
| $\quad$ **for** $e_i \in E_a$ **do** | $\quad$ **for** $e_i \in E_a$ **do** |
| $\qquad c_{i,t+1} := c_{i,t} \cdot e^{-\eta \ell_{i,t}}$ | $\qquad c_{i,t+1} := c_{i,t} \cdot e^{-\eta \ell_{i,t}}$ |
| $\qquad h_{i,t+1} := h_{i,t}$ | $\qquad h_{i,t+1} := h_{i,t}$ |
| $\quad$ **if** *expert j dies* **then** | $\quad$ **if** *expert j dies* **then** |
| $\qquad E_a := E_a \setminus \{e_j\}$ | $\qquad E_a := E_a \setminus \{e_j\}$ |
| $\qquad$ **for** $e_i \in E_a$ **do** | $\qquad$ **for** *each* $i = j+1$ *to* $K$ **do** |
| $\qquad\quad h_{i,t+1} := h_{i,t+1} \cdot c_{i,t+1}$ | $\qquad\quad h_{i,t+1} := h_{i,t+1} \cdot c_{i,t+1}$ |
| $\qquad\qquad + (h_{j,t+1} \cdot c_{j,t+1})/|E_a|$ | $\qquad\qquad + (h_{j,t+1} \cdot c_{j,t+1})\left(\lceil 2^{i-2}\rceil / 2^{K-1-j}\right)$ |
| $\qquad\quad c_{i,t+1} := 1$ | |
| | $\qquad\quad c_{i,t+1} := 1$ |

algorithm works, after which we prove that Algorithm 1 efficiently simulates running Hedge over $\Pi$. Before proceeding further, let us recall how Hedge makes predictions in round $t$. First, it updates the weights using $w_{i,t} = w_{i,t-1} e^{-\eta \ell_{i,t}}$, and it then assigns a probability to expert $i$ as follows:

$$p_{i,t} = \frac{w_{i,t-1}}{\sum_{i=1}^{K} w_{i,t-1}} \ .$$

Recall that $e_1, e_2, \ldots, e_K$ denote the original experts while $\pi_1, \pi_2, \ldots \pi_{K!}$ denote the orderings. Denote by $w_\pi^t$ the weight that Hedge assigns to $\pi$ in round $t$. Define $\Pi_i^t \subseteq \Pi$ to be the set of orderings predicting as expert $e_i$ in round $t$. The main ideas behind the algorithm are as follows:

1. When $\pi$ and $\pi'$ have the same prediction $e$ in round $t$ (i.e. $\sigma^t(\pi) = \sigma^t(\pi') = e$), then we do not need to know $w_\pi^t$ and $w_{\pi'}^t$; we use $w_\pi^t + w_{\pi'}^t$ instead for the weight of $e$.

2. The algorithm maintains $\sum_{\pi \in \Pi_j^t} e^{-\eta L_\pi^{t-1}}$, where $\eta$ is the learning rate and $L_\pi^t$ is the cumulative loss of ordering $\pi$ up until round $t$, i.e., $L_\pi^t = \sum_{s=1}^{t} \ell_{\sigma^s(\pi),s}$.

We will discuss how to tune $\eta$ later. Let $J = \{j_1, \ldots, j_m\}$ represent the rounds on which any expert will die. Denote by $j_t$ the last night observed so far at the end of round $t$, formally defined as $j_t = \max_{j \in J} j \le t$. We maintain a tuple $(h_{i,t}, c_{i,t})$ for each original expert $e_i$'s in the algorithm in round $t$, where $h_{i,t}$ is the sum of non-normalized weights of the experts in $\Pi_i^t$ in round $j_t$. We similarly maintain $c_{i,t}$, except that it only considers the loss suffered from $j_t + 1$ to round $t-1$ for experts in $\Pi_i^t$. Formally:

$$h_{i,t} = \sum_{\pi \in \Pi_i^t} e^{-\eta(\sum_{s=1}^{j_t} \ell_{\sigma^s(\pi),s})}, \qquad c_{i,t} = \sum_{\pi \in \Pi_i^t} e^{-\eta(\sum_{s=j_t+1}^{t-1} \ell_{\sigma^s(\pi),s})} \ .$$

It is easy to verify that $h_{j,t} \cdot c_{j,t} = \sum_{\pi \in \Pi_j^t} e^{-\eta L_\pi^{t-1}}$.

The computational cost of the algorithm at each round will be $\mathcal{O}(K)$. We claim that HPU will behave the same as executing Hedge on $\Pi$. We use induction on rounds to show the weights are the same in both algorithms. By "simulating" we mean that the weights over the original experts will be maintained identically to how Hedge maintains them.

**Theorem 5.1.** *At every round, HPU simulates running Hedge on the set of experts $\Pi$.*

*Proof Sketch.* The main idea is to group the permutation experts with similar predictions (the first expert alive in the permutation) in one group. Hence, initially there will be $K$ groups. Then, if expert $e_j$ dies, every ordering in the group associated with $e_j$ will be moved to another group and the empty group will be deleted. We prove that the orderings will distribute to other groups symmetrically after a night. Using this fact, we show that we do not need to know the elements of a group; we only maintain the sum of the weights given to all the orderings in each group. $\qquad\square$

## 5.2 Known order of dying

For the case of known order of dying, we propose Algorithm 2, Hedge-Perm-Known (HPK), which is slightly different than HPU. In particular, the weight redistribution (when an expert dies) and initialization of coefficient $h_{i,1}$ is different. In the proof of Theorem 5.1, we showed that when the set of experts includes all the orderings, the weight of the expert $j$ that died will distribute equally between initial experts ($e_j \in E$). But when the set of experts is only the effective experts, this no longer holds. In this section, we assume without loss of generality that the experts die in the order $e_1, e_2, \ldots$ and recall that $\mathcal{E}$ denotes the set of effective orderings. Based on Theorem 3.1, the number of experts starting with $e_i$ in $\mathcal{E}$ is $\lceil 2^{K-i-1} \rceil$; we denote the set of such experts as $\mathcal{E}_{e_i}$.

**Theorem 5.2.** *At every round, HPK simulates running Hedge on the set of experts $\mathcal{E}$.*

**Remarks for tuning learning rates.**     For both algorithms, we assume $T$ is known beforehand. So, the learning rate for HPU is $\eta = \sqrt{(2 \log(K!))/T}$ and for HPK is $\eta = \sqrt{(2 \log(2^m(K-m)))/T}$. One can use a time-varying learning rate to adapt to $T$ in case it is not known.

## 5.3 Extensions for algorithms

As we mentioned at the beginning of Section 5, for the sake of simplicity we initially assumed that only one expert dies each night. First, we discuss how to handle a night with more than one death. Afterwards, we explain how to extend/modify HPU and HPK to implement the Follow The Leader (FTL) strategy. We then introduce a new algorithm which simulates FTL efficiently and maintains $L_t^*$ as well, where $L_t^*$ is the cumulative loss of the best permutation expert through the end of round $t$. Finally, using $L_t^*$, we explain how to simulate AdaHedge and FlipFlop [5] by slightly extending HPU and HPK.

**More than one expert dying in a night.**     We handle nights with more than one death as follows. We have one of the experts die on that night, and, for each expert $j$ among the other experts that should have died that night, we create a "dummy round", give all alive experts (including expert $j$) a loss of zero, keep the learning rate the same as the previous round, and have expert $j$ die at the end of the dummy round (which hence becomes a "dummy night"). Even though the number of rounds increases with this trick, it is easy to see that the regret is unchanged since in dummy rounds all experts have the same loss (and also the learning rate after the sequence of dummy rounds is the same as what it would have been had there been no dummy rounds). Moreover, since now one expert dies on each night (some of which may be dummy nights), we may use Theorems 5.1 or 5.2 to conclude that our algorithm correctly distributes any dying experts' weights among the alive experts.

**Beyond adaptivity to m.**     Consider the case of unknown order and let the number of nights $m$ be unknown. As promised, we show that we can improve on the simple Resetting-Hedge strategy.

**Theorem 5.3.** *Consider running Hedge on top of $K$ copies of HPU where, for $r \in \{0, 1, \ldots, K-1\}$, we set $\varepsilon_r = \prod_{l=0}^{r-1} \frac{1}{K-l}$ and the $r^{th}$ copy of HPU uses learning rate $\eta_t^{\varepsilon_r} := \sqrt{8 \log(1/\varepsilon_r)/t}$. Let $\pi^*$ be a best permutation expert in hindsight and suppose that the sequence $(\sigma^1(\pi^*), \ldots, \sigma^T(\pi^*))$ changes experts at most $l$ times. Then the regret of this algorithm is $\mathcal{O}\big(\sqrt{T(l+1)\log K}\big)$.*

Note that this theorem does better than adapt to $m$, as with $m$ nights we always have $l \leq m$ but $l$ can in fact be much smaller than $m$ in practice. Hence, Theorem 5.3 recovers and can improve upon the regret of Resetting-Hedge and, moreover, wasteful resetting is avoided. Also, while the computation increases by a factor of $K$, it is easy to see that one can instead use an exponentially spaced grid of size $\log_2(K)$ to achieve regret of the same order.

**Follow the Leader.**     FTL might be the most natural algorithm proposed in online learning. In round $t$ the algorithm plays the expert with the lowest cumulative loss up to round $t$, $L_{t-1}^*$. By setting $\eta = \infty$ in Hedge and similarly, in HPU and HPK, we recover FTL; hence, our algorithms can simulate FTL. The motivation for FTL is that it achieves constant regret (with respect to $T$) when the losses are i.i.d. stochastic and there is a gap in mean loss between the best and second best (permutation) experts. Our algorithms do not maintain $L_t^*$, but we need $L_t^*$ to implement AdaHedge (which we discuss in the next extension). Here, we propose a simple algorithm to perform FTL on the set of orderings. The algorithm works as follows:

1. Perform as FTL on alive initial experts and keep track of their cumulative losses $(L_1^t, L_2^t, \ldots, L_K^t)$, while ignoring the dead experts;

2. If expert $j$ dies in round $t'$, then for every alive expert $i$ where $L_i^{t'} > L_j^{t'}$ do: $L_i^{t'} := L_j^{t'}$.

This not only performs the same as FTL but also explicitly keeps track of $L_t^*$. We will use this implementation to simulate AdaHedge.

**AdaHedge.** The following change to the learning rate in HPU/HPK recovers AdaHedge. Let $\hat{L}_t = \sum_{r=1}^t \hat{\ell}_r$. For round $t$, AdaHedge on $N$ experts sets the learning rate as $\eta_t = (\ln N)/\Delta_{t-1}$ and $\Delta_t = \hat{L}_t - M_t$ where $M_t = \sum_{r=1}^t m_r$ and $m_r = -\frac{1}{\eta_r} \ln(\boldsymbol{w_r} \cdot e^{-\eta_r \boldsymbol{\ell}_r})$; here, $m_r$ can easily be computed using the weights from HPU/HPK. As we have the loss of the algorithm at each round, we can calculate $M_t$. Also, using the implementation of FTL describe above, we can maintain $L_t^*$. Finally, we can compute $\Delta_t$ and the regret of HPU/HPK.

**FlipFlop.** By combining AdaHedge and FTL, [5] proposes FlipFlop which can do as well as either of AdaHedge (minimax guarantees and more) or FTL (for the stochastic i.i.d. case). We can adapt HPK and HPU to FlipFlop by implementing AdaHedge and FTL as described above and switching between the two based on $\Delta_t^{\mathrm{ah}}$ and $\Delta_t^{\mathrm{ftl}}$, where $\Delta_t^{\mathrm{ftl}}$ is defined similar to $\Delta_t^{\mathrm{ah}}$ but the learning rate associated with $m_t$ for FTL is $\eta_t^{\mathrm{ftl}} = \infty$ while for AdaHedge it is $\eta_t^{\mathrm{ah}} = \frac{\ln K}{\Delta_{t-1}}$.

**Corollary 5.1.** *By combining FTL and AdaHedge as described above, HPU and HPK simulate FlipFlop over set of experts $A$ (where $A = \Pi$ for HPU and $A = \mathcal{E}$ for HPK) and achieve regret*

$$R_A(1,T) < \min\left\{ C_0 R_A^{\mathrm{ftl}}(1,T) + C_1, C_2 \sqrt{\frac{L_T^*(T - L_T^*)}{T} \ln(|A|)} + C_3 \ln(|A|) \right\},$$

*where $C_0, C_1, C_2, C_3$ are constants.*

The interest in FlipFlop is that in the real-world we may not know if losses are stochastic or adversarial. This motivates one to use an algorithms that detect and adapt to easier situations.

# 6   Conclusion

In this work, we introduced the dying experts setting. We presented matching upper and lower bounds on the ranking regret for both the cases of known and unknown order of dying. In the case of known order, we saw that the reduction in the number of effective orderings allows our bounds to be reduced by a $\sqrt{\log K}$ factor. While it appears to be computationally hard to obtain sublinear regret in the general sleeping experts problem, in the restricted dying experts setting we provided efficient algorithms with optimal regret bounds for both cases. Furthermore, we proposed an efficient implementation of FTL for dying experts which, combined with efficiently maintaining mix losses, enabled us to extend our algorithms to simulate AdaHedge and FlipFlop. It would be interesting to see if the notion of effective experts can be extended to other settings such as multi-armed bandits. Furthermore, it might be interesting to study the problem in regimes in between known and unknown order.

### Acknowledgments

This work was supported by the NSERC Discovery Grant RGPIN-2018-03942.

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
