[Supplementary Material]

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

# A Proofs for Section 3

## A.1 Proof of Theorem 3.1

We define a new operator denoted by $+$ which operates between an expert $e$ on the left hand side and a multi-set of orderings $\Pi$ on the right hand side and returns a new multi-set of orderings in which $e$ is added to the left side of every ordering $\pi \in \Pi$. Let $J = \{j_1, \ldots, j_m\}$ be the rounds on which any expert will die.

*Proof.* Without loss of generality, assume that experts die in the order of their indices, i.e., $e_1$ dies first, $e_2$ second, ..., and $d_{K-A}$ dies last. We use mathematical induction on $m$ to prove the claim.

*Induction Basis:* For $m = 0$, or the case that no expert dies (i.e. $A = K$), the number of effective permutation experts is equal to $A$, the number of experts that never die. Hence,

$$f(\{\}, A) = A.$$

*Induction Hypothesis:* We assume that the number of effective experts when there are only $i - 1$ nights is equal to

$$f(\{d_1, d_2, \ldots, d_{i-1}\}, A) = A \prod_{s=1}^{i-1} (d_s + 1)$$

Denote the set of the effective permutations created in the induction hypothesis as $\mathcal{E}_{i-1}$.

*Induction Step:* First, notice that any expert $e \in E_d^{j_1+1}$ has an impact on the behavior of a permutation expert $\pi$ only if $e = \sigma^1(\pi)$. If we ignore the first night and remove every $e \in E_d^{j_1+1}$ from the orderings, the theorem would behave as though those experts do not exist and there are only $i - 1$ nights. Due to the induction hypothesis we know that

$$f(\{d_2, \ldots, d_i\}, A) = A \prod_{s=2}^{i} (d_s + 1).$$

Denote by $F$ the number of effective orderings $\pi$ where $e = \sigma^1(\pi)$ for some $e \in E_d^{j_1+1}$. It is easy to see that

$$f(\{d_1, d_2, \ldots, d_i\}, A) = F + f(\{d_2, \ldots, d_i\}, A).$$

On the other hand, the effective orderings which start with $e_s \in E_d^{j_1+1}$ can be constructed as $(e_s) + \mathcal{E}_{i-1}$, so

$$|(e_s) + \mathcal{E}_{i-1}| = |\mathcal{E}_{i-1}| = f(\{d_2, \ldots, d_i\}, A),$$

and it follows that $\mathcal{E}_i = (\cup_{e_s \in E_d^{j_1+1}} ((e_s) + \mathcal{E}_{i-1})) \cup \mathcal{E}_{i-1}$. Then due to the induction hypothesis we get:

$$f(\{d_1, d_2, \ldots, d_i\}, A) = (d_1 + 1) f(\{d_2, \ldots, d_i\}, A) = (A)\Pi_{s=1}^{i}(d_s + 1) \tag{5}$$

and (5) completes the induction step, concluding the proof. □

# B Proofs for Section 4

## B.1 Proof of Theorem 4.1

Our lower bound strategy is similar[2] to the proof of Theorem 3.7 of [3] until our equation (6).

*Proof.* Any strategy of Learner over $T$ rounds can be represented as a sequence $\boldsymbol{p}$ of $T$ maps $\boldsymbol{p}_1, \ldots, \boldsymbol{p}_T$, where

$$\boldsymbol{p}_t : [0, 1]^{t-1} \to \Delta_K.$$

By representing Learner's strategy in this way, we can write the above minimax regret as:

$$\inf_{\mathbf{P}} \sup_{\boldsymbol{\ell}_1,\dots,\boldsymbol{\ell}_T} \left\{ \sum_{t=1}^T \boldsymbol{p}_t \cdot \boldsymbol{\ell}_t - \min_{j \in [K]} \sum_{t=1}^T \ell_{j,t} \right\}.$$

The minimax regret can only decrease by replacing the supremum over the experts' losses by an expectation over random i.i.d. losses $\ell_{j,t}$, where, for all $j \in [K]$ and $t \in [T]$, we take $\ell_{j,t}$ to be independently drawn uniformly from $\{0,1\}$; hence, the above is lower bounded by

$$\inf_{\mathbf{P}} \mathsf{E}\left[ \sum_{t=1}^T \boldsymbol{p}_t \cdot \boldsymbol{\ell}_t - \min_{j \in [K]} \sum_{t=1}^T \ell_{j,t} \right] = \mathsf{E}\left[ \frac{T}{2} - \min_{j \in [K]} \sum_{t=1}^T \ell_{j,t} \right]$$

$$= \frac{1}{2} \mathsf{E}\left[ \max_{j \in [K]} \sum_{t=1}^T (1 - 2\ell_{j,t}) \right].$$

Now, observe that each random variable $(1 - 2\ell_{j,t})$ has the same law as an independent Rademacher random variable (i.e. uniform over $\pm 1$). Therefore, the above is equal to

$$\frac{1}{2} \mathsf{E}\left[ \max_{j \in [K]} \sum_{t=1}^T \varepsilon_{j,t} \right], \tag{6}$$

where the $\varepsilon_{j,t}$ are independent Rademacher random variables.

Our approach will be to express the above as the expected maximum of a Bernoulli process. To this end, for each $j \in [K]$ define the matrix $\tau^{(j)} \in \mathbb{R}^{T \times K}$ whose $j^{\text{th}}$ column is equal to the ones vector and whose remaining columns each are equal to the zero vector. Our lower bound on the minimax regret now can be rewritten as (one half of)

$$\mathsf{E}\left[ \max_{j \in [K]} \sum_{t=1}^T \varepsilon_{j,t} \right] = \mathsf{E}\left[ \max_{j \in [K]} \sum_{i=1}^K \sum_{t=1}^T \varepsilon_{i,t} \tau_{i,t}^{(j)} \right].$$

This is just the expected supremum of a Bernoulli process indexed by $\{\tau^{(1)}, \dots, \tau^{(K)}\}$. A result of [18], restated as Lemma B.1 after this proof, can be used to lower bound this process in terms of $T$ and $K$. In our setting, treating the matrices $\tau^{(j)}$ as vectors by stacking their columns, we see that the vectors $(\tau^{(j)})_j$ satisfy

- $\|\tau^{(i)} - \tau^{(j)}\|_2 \geq \sqrt{2T}$ for all distinct $i, j \in [K]$;

- $\|\tau^{(j)}\|_\infty \leq 1$ .

Hence, Lemma B.1 implies that the minimax regret is lower bounded by

$$\frac{1}{2L} \min\left\{ \sqrt{2T \log K}, 2T \right\},$$

for $L$ a universal constant. $\qquad\square$

The above proof uses the following powerful result of Talagrand on Sudakov minoration for Bernoulli processes; here, the most convenient form is stated as Theorem 4.2.4 in [18], but the result first appeared as Proposition 2.2 of [17] in a quite different form.

**Lemma B.1** (Sudakov minoration for Bernoulli processes [18, Theorem 4.2.4])**.** *Let $a, b > 0$ and $\tau^{(1)}, \dots, \tau^{(K)} \in \ell^2$ satisfy the conditions:*

- $\|\tau^{(i)} - \tau^{(j)}\|_2 \geq a$ *for all distinct $i, j \in [K]$ ;*

- $\|\tau^{(j)}\|_\infty \leq b$ *for all $j \in [K]$ ,*

*Let $\varepsilon_1, \varepsilon_2, \dots$ be i.i.d Rademacher random variables.*

*Then for a universal constant $L$ we have*

$$\mathsf{E} \sup_{j \leq K} \sum_{s \geq 1} \tau_s^{(j)} \varepsilon_s \geq \frac{1}{L} \min\left\{ a\sqrt{\log K}, \frac{a^2}{b} \right\}.$$

## B.2  Proof of Theorem 4.2

*Proof.* We prove the theorem using a construction as follows. Recall that we refer to a round as a "night" if an expert dies on that round and to each segment between two nights as a "day". First, partition $T$ rounds to rounds of length $T' = T/(m+1)$, where $m$ is the number of nights. The goal is to construct a scenario where each day is a game decoupled from the previous ones. This means that the algorithm will be forced to have no prior information about the experts at the beginning of each day. Recall that $\tau_s$ is the set of time-step indices of day $s$, i.e.,$\tau_s = \{t | (s-1)T' < t \le sT'\}$.

Each day is divided into two equal parts. Denote by $\tau_s^1$ and $\tau_s^2$ sets of time-step indices of the first half and the second half of day $s$, respectively. Let $\boldsymbol{\ell}_{\tau_s^1, i}$ and $\boldsymbol{\ell}_{\tau_s^2, i}$ be the sequences of losses of an expert $i$ on the first and second half of the day $s$, respectively. On the first half of day $s$, each expert suffers loss drawn i.i.d. from a Bernoulli distribution with $p = 1/2$. At the end of the first half of the day, we choose the expert with the lowest cumulative loss up until now, denoted by $e_s^*$. This expert will suffer no loss in the second half. Also, the adversary forces every other expert $i$ where $e_i \ne e_s^*$ to suffer losses according to loss sequence $\boldsymbol{\ell}_{s_2, i}$ on the second half of the day, where we have element-wise subtraction as $\boldsymbol{\ell}_{s_2, i} = \mathbf{1} - \boldsymbol{\ell}_{s_1, i}$. Denote by $E_a(s)$ the set of experts alive on day $s$.

We now analyze the ranking regret of any algorithm for this construction over $T$ rounds. Without loss of generality, suppose the order of experts that are going to die is as $\mathcal{D} = (e_1, e_2, \ldots, e_m)$. Also, denote by $\pi^* \in \Pi$ the best ordering over $T$. From the construction, it is clear that $\pi^* = (\mathcal{D}, \ldots)$. Therefore, the ranking regret over $T$ rounds is obtained from (7).

$$R_\Pi(1, T) = \hat{L} - L_{\pi^*} = \hat{L} - \sum_{s=1}^{m+1} \sum_{t \in \tau_s} l_{\sigma^t(\pi^*), t} \tag{7}$$

where $L_{\pi^*}$ is the cumulative loss of playing according to ordering $\pi^*$. Now we write $R_\Pi(1, T)$ in terms of a sum of classical regrets over each day. Since in our construction the best expert of the day will die at the end of that day, then, for all rounds in a given day, $\sigma^t(\pi^*)$ yields the same expert as the expert that is best for that day according to the ordinary regret. Therefore, we have:

$$\hat{L} - \sum_{s=1}^{m+1} \sum_{t \in \tau_s} l_{\sigma^t(\pi^*), t} = \sum_{s=1}^{m+1} \left( \sum_{t \in \tau_s} \hat{l}_t - \min_{s \le i \le K} \sum_{t \in \tau_s} l_{i, t} \right)$$

$$= \sum_{s=1}^{m+1} R_{E_a(s)}(\tau_s) \tag{8}$$

where the last equality is obtained from the fact that in our construction, each day is an independent day from the others, meaning the history of losses of the experts does not matter. Combining (8) and (7), we have:

$$R_\Pi(1, T) = \sum_{s=1}^{m+1} R_{E_a(s)}(\tau_s) \tag{9}$$

Now it remains to analyze the regret of each day separately. For this, first, we lower bound the regret of each half of the days. Denote by $\hat{L}_s^1$ and $\hat{L}_s^2$ the cumulative losses of the algorithm on the first and second half of the day $s$ with length, respectively. It is easy to verify that $R_{E_a(s)}(\tau_s) \ge R_{E_a(s)}(\tau_s^2)$, hence for the regret of each day we have

$$R_{E_a(s)}(\tau_s) = \hat{L}_s^1 + \hat{L}_s^2 - \sum_{t \in \tau_s} l_{\sigma^t(\pi^*), t} \ge \hat{L}_s^1 - \sum_{t \in \tau_s^1} l_{\sigma^t(\pi^*), t}$$

$$\ge \frac{1}{L} \min\{\sqrt{T'/2 \log(K - s)}, T'\} \tag{10}$$

where the last inequality is based on (the proof of) Theorem 4.1. Combining (9) and (10) we have:

$$R_\Pi(1, T) \ge \sum_{s=1}^{m+1} \frac{1}{L} \min\{\sqrt{T'/2 \log(K - s)}, T'\}$$

$$= \sum_{s=1}^{m+1} \sqrt{T/2(m+1) \log(K - s)} = \Omega\left(\sqrt{Tm \log K}\right), \tag{11}$$

yielding the desired bound. □

The proof of Theorem 4.2 uses the results of following Lemma.

**Lemma B.2.** *For variables $x_1, \ldots, x_m > 0$ and subject to $\sum_{i=1}^{m} x_i = T$, we have:*

$$\sum_{i=1}^{m} \sqrt{x_i} \leq \sqrt{mT}$$

*Proof.* Denote by $\mathcal{T}$ the vector $\left[\sqrt{x_1}, \sqrt{x_2}, \ldots, \sqrt{x_m}\right]$ and $I = [1, 1, \ldots, 1]$ vectors of length $m$ where all the elements are equal to one. We have:

$$\sum_{i=1}^{m} \sqrt{x_i} = \mathcal{T} \cdot I^T \leq ||\mathcal{T}|| \cdot ||I|| \leq \sqrt{m}\sqrt{(\sqrt{x_1})^2 + (\sqrt{x_2})^2 + \cdots + (\sqrt{x_m})^2} \leq \sqrt{mT}$$

where the first inequality follows from the Cauchy-Schwarz inequality. $\square$

### B.3 Proof of Theorem 4.4

*Proof.* The construction for this case is similar to the one we previously had for the unknown order of dying. We divide the $T$ rounds into $m/2$ days each of size $T' = 2T/m$. We choose two experts $\{e_{2s-1}, e_{2s}\}$ at each day $s$ and they will suffer losses drawn i.i.d. from a Bernoulli distribution with success probability of $p = 1/2$. Every expert $e_i \notin \{e_{2s-1}, e_{2s}\}$ will suffer constant loss of 1 during day $s$. At the end of day $s$, both the experts $\{e_{2s-1}, e_{2s}\}$ will die. Therefore, at the beginning of each day, all the experts have the same loss history and consequently, each day is decoupled from the previous ones. Additionally, we provide extra information to the algorithm, that the best expert of day $s$ is one of the two experts $\{e_{2s-1}, e_{2s}\}$. Thus, the algorithm needs to track only two experts on a single day.

Denote by $E_a(s)$ the set of experts alive on day $s$. In the following, we analyze the ranking regret of this construction. We will use the same result from the proof of Theorem 4.2 to connect ranking regret to the classical regret over each day. Hence, using (9), we have:

$$R_\Pi(1, T) = \sum_{s=1}^{m/2} R_{E_a(s)}(\tau_s) \tag{12}$$

Using the bound we obtained from Theorem 4.1, for each day $s$, $K = 2$ and $T'$ rounds we have:

$$R_{E_a(s)}(\tau_s) \geq \frac{1}{L}\min\{\sqrt{T'/2\log 2}, T'\} \tag{13}$$

Combining (12) and (13), we obtain the bound on ranking regret over time horizon $T$ as follows:

$$R_\Pi(1, T) \geq \sum_{s=1}^{m/2} \frac{1}{L}\min\{\sqrt{T'/2\log 2}, T'\} = \sum_{s=1}^{m/2} \sqrt{T/m} = \Omega\left(\sqrt{mT}\right)$$

The theorem follows. $\square$

## C Proofs for Section 5

Wherever we refer to Theorem 3.1 in this section, we assume that only one expert dies each night; therefore for $m$ nights (consequently, $m$ dying experts) the value of $f$ (the function defined in Theorem 3.1) is $2^m(K - m)$ where $K$ is the number of experts.

### C.1 Proof of Theorem 5.1

*Proof.* We show that the loss and weights of the algorithms are the same at each round, therefore, their regret is the same. Define $\Pi_D$ to be the set of all possible orderings of elements in set $D$ of length $|D|$. We claim that based on the update rules of HPU for $h_{j,t}$ and $c_{j,t}$, for every round and expert we have $\sum_{\pi \in \Pi_j^t} e^{-\eta L_\pi^{t-1}} = h_{j,t} \cdot c_{j,t}$.

*Induction Basis*: At round $t = 1$, in Hedge, every expert has non-normalized weight of 1. The size of each $\Pi_{e_j}^1$ is $(K-1)!$. The algorithm assigns $h_{i,1} = (K-1)!$ and $c_{i,1} = 1$, therefore the claims hold.

*Induction Hypothesis*: At the beginning of round $t-1$, for every alive expert $e_j$, the following holds:

$$\sum_{\pi \in \Pi_j^{t-1}} e^{-\eta L_\pi^{t-2}} = h_{j,t-1} \cdot c_{j,t-1} \tag{14}$$

*Induction Step*: This step is divided into two cases. First, when $E_a^t = E_a^{t-1}$. Second, when an expert dies, $|E_a^t| = |E_a^{t-1}| - 1$.

*Case I*: If no expert dies at the end of round $t-1$, then for every $i$ we have $\Pi_i^t = \Pi_i^{t-1}$ and $h_{i,t} = h_{i,t-1}$, thus for every alive expert $e_j$ following holds $\sum_{\pi \in \Pi_j^t} e^{-\eta L_\pi^{t-2}} = h_{j,t-1} \cdot c_{j,t-1}$. After the update in Hedge, the weights on $\Pi_j^t$ is $\sum_{\pi \in \Pi_j^t} e^{-\eta(L_\pi^{t-2} + \ell_{j,t-1})}$. On the other hand, in the HPU algorithm, we have the following:

$$h_{j,t} \cdot c_{j,t} = e^{-\eta \ell_{j,t-1}} \cdot c_{j,t-1} \cdot h_{j,t-1} = e^{-\eta \ell_{j,t-1}} \sum_{\pi \in \Pi_j^{t-1}} e^{-\eta L_\pi^{t-2}}$$

$$= \sum_{\pi \in \Pi_j^t} e^{-\eta(L_\pi^{t-2} + \ell_{j,t-1})} = \sum_{\pi \in \Pi_j^t} e^{-\eta L_\pi^{t-1}}$$

Where the second equality follows from the induction hypothesis. It can be observed that the weights are identical to the ones from running Hedge on $K!$ experts.

*Case II*: The second case is when the expert $j$ dies at the end of round $t-1$. Let $i, k$ be arbitrary alive experts not equal to $j$. Observe that any $\pi \in \Pi_i^t \cap \Pi_j^{t-1}$ takes the form $(\pi_d, e_j, \pi_{d'}, e_i, \pi_{R_i})$, where, for some $D, D' \subseteq E_d^t$ where $D \cap D' = \emptyset$ and $R_i := E \setminus (D \cup D' \cup \{e_j, e_i\})$, we have that $\pi_d \in \Pi_D$ and $\pi_{d'} \in \Pi'_D$ and $\pi_{R_i} \in \Pi_{R_i}$. Then $\Pi_k^t \cap \Pi_j^{t-1}$ contains a unique element $\pi' = (\pi_d, e_j, \pi_{d'}, e_k, \pi_{R_k})$, where D is taken as before and (like before) $R_k := E \setminus (D \cup D' \cup \{e_j, e_k\})$ and $\pi_{R_k}$ is created only by replacing $e_i$ as $e_k$ in $\pi_{R_i}$. Moreover, since their behavior is the same over the first $t-1$ rounds, $\pi$ and $\pi'$ satisfy $L_\pi^{t-1} = L_{\pi'}^{t-1}$.

Therefore by symmetry, we can obtain (16) from (15).

$$\sum_{\pi \in \Pi_i^t} e^{-\eta L_\pi^{t-1}} = \sum_{\pi \in \Pi_i^{t-1}} e^{-\eta L_\pi^{t-1}} + \sum_{\pi \in (\Pi_i^t \cap \Pi_j^{t-1})} e^{-\eta L_\pi^{t-1}} \tag{15}$$

$$= \sum_{\pi \in \Pi_i^{t-1}} e^{-\eta L_\pi^{t-1}} + \frac{1}{|E_a^t|} \left( \sum_{\pi \in \Pi_j^{t-1}} e^{-\eta L_\pi^{t-1}} \right) \tag{16}$$

$$= h_{i,t} \cdot c_{i,t}$$

Notice that, given expert $j$ dies at the end of round $t-1$ hence, $\sum_{\pi \in \Pi_j^{t-1}} e^{-\eta L_\pi^{t-1}}$ is calculable. Therefore, HPU is always maintaining the weights correctly. $\square$

## C.2 Proof of Theorem 5.2

Here we follow a construction similar to the proof of Theorem 5.1, i.e., we do induction on $t$. Before proceeding to the proof, define $\lambda(\pi, t)$ as a function that will remove ineffective elements of a permutation expert at round $t$. An element is said to be ineffective, if it will never be used for the prediction in that permutation or it is dead. Recall that in this section we assumed that the experts die in order, $e_1$ dies first and $e_{K-A}$ last. For example, $(e_4) = \lambda((e_4, e_3, e_2, e_1), t)$ and $(e_1, e_3, e_4) = \lambda((e_1, e_3, e_2, e_4), 1)$ with respect to the assumption we made earlier on the order of dying, and if $e_1$ dies at $t = 1$ and $e_3$ dies at $t = 5$, then $(e_3, e_4) = \lambda((e_1, e_3, e_2, e_4), 3)$ and $(e_4) = \lambda((e_1, e_3, e_2, e_4), 6)$. Naturally, $\lambda(\mathcal{E}, t)$ performs the function $\lambda(\pi, t)$ on every permutation $\pi \in \mathcal{E}$. The output of $\lambda(\mathcal{E}, t)$ is a multi-set, not a set.

*Proof. Induction Basis*: At round $t = 0$, each of the permutation-experts have the same weight. Due to the Theorem 3.1, we know the number of the orderings starting by expert $e_i$ is equal to $\lceil 2^{K-i-1} \rceil$.

Therefore, in Hedge, the cumulative non-normalized weight put on $\mathcal{E}_{e_i}^1$ is $\lceil 2^{K-i-1} \rceil$ which is equal to $h_{i,1} \cdot c_{i,1}$ in HPK.

*Induction Hypothesis*: At the beginning of round $t-1$, in Hedge, the cumulative non-normalized weight put on $\mathcal{E}_{e_i}^{t-1}$ for every $e_i \in E_a^{t-1}$, is equal to $h_{i,t-1} \cdot c_{i,t-1}$

*Induction Step*: As in the proof of Theorem 5.1, for round $t$, we split the step into two cases. In the first case, no expert dies, i.e., $E_a^t = E_a^{t-1}$. For the second case, one expert dies at the end of round $t-1$. The proof for the first case is omitted as it is identical to the proof for *Case I* of Theorem 5.1. For the case that an expert dies at the end of round $t-1$, we show that the weight distribution works correctly.

Let $E^{(i+1,K)} = \{e_{i+1}, \ldots, e_K\}$. Due to the discussions in Section 3, first, we have the number of initial orderings starting by $e_i$ as $\lceil 2^{K-i-1} \rceil$ at the beginning. Second, due to Lemma C.1, if $e_i$ dies at round $t-1$, we have:

$$\lambda\left(\mathcal{E}_{e_i}^{t-1}, \ t\right) = \lambda\left(\bigcup_{e \in E^{(i+1,K)}} \mathcal{E}_e^{t-1}, \ t\right) \tag{17}$$

therefore, for every $e_j$ where $j > i$, $(|\mathcal{E}_{e_j}^1|)/(|\mathcal{E}_{e_i}^1|)$ fraction of $h_{i,t-1} \cdot c_{i,t-1}$ must be added to the weight of $\mathcal{E}_{e_j}^t$ to maintain the weight of $\mathcal{E}_{e_j}^t$. □

Before proceeding to Lemma C.1, recall that operator $+$ operates between an expert $e$ on LHS and a multi-set of orderings $\Pi$ on RHS and returns a new multi-set of orderings which $e$ is added to the left side of every ordering $\pi \in \Pi$.

**Lemma C.1.** *At round $t$, where $E_d^t = \{e_1, e_2, \ldots e_{i-1}\}$ are dead and the rest of the experts are alive, we have:*

$$\lambda\left(\mathcal{E}_{e_i}^t, \ t\right) = (e_i) + \lambda\left(\bigcup_{e \in E^{(i+1,K)}} \mathcal{E}_e^t, \ t\right)$$

*and therefore:*

$$|\mathcal{E}_{e_i}^t| = \left|\lambda\left(\bigcup_{e \in E^{(i+1,K)}} \mathcal{E}_e^t, \ t\right)\right|.$$

Before proving the statement, let us define two new operators. For $\mathcal{E}$ as the set of permutation-experts, $\mathcal{E} - \{e_i\}$ removes element $e_i$ from every permutation $\pi \in \mathcal{E}$. Also, $\mathcal{E}' = x\mathcal{E}$ is a multi-set where each item in $\mathcal{E}$ is copied $x$ times. As a result, trivially we have $|\mathcal{E}'| = x \cdot |\mathcal{E}|$.

*Proof.* Recall that we assumed that the experts die in order. Due to constructive structure of the Theorem 3.1, $\mathcal{E}_{e_i}^1$ is equal to adding $e_i$ as the first element for every permutation in $\mathcal{E}_{E^{(i+1,K)}}^1$.

$$\lambda\left((\mathcal{E}_{e_i}^1), \ 1\right) = (e_i) + \lambda\left(\bigcup_{e \in E^{(i+1,K)}} \mathcal{E}_e^1, \ 1\right)$$

Therefore, the claim holds for $t=1$ and we have $|\lambda(\mathcal{E}_{e_i}^1, 1)| = |\mathcal{E}_{e_i}^1|$. It is easy to verify that:

$$\left|\lambda\left(\bigcup_{e \in E^{(i+1,K)}} \mathcal{E}_e^1, \ 1\right)\right| = \left|\bigcup_{e \in E^{(i+1,K)}} \mathcal{E}_e^1\right|$$

Due to Lemma C.2, similar claim holds for $t$ when $\{e_1, \ldots, e_{i-1}\}$ are dead. $\lambda(\mathcal{E}_{e_i}^t, t)$ will be $2^{i-1}$ copies of $\lambda(\mathcal{E}_{e_i}^1, 1)$ and similarly

$$\lambda\left(\bigcup_{e \in E^{(i+1,K)}} \mathcal{E}_e^t, \ t\right) = 2^{i-1}\lambda\left(\bigcup_{e \in E^{(i+1,K)}} \mathcal{E}_e^1, \ 1\right)$$

hence:

$$2^{i-1}\lambda\left(\mathcal{E}_{e_i}^1,\ 1\right) = (e_i) + 2^{i-1}\lambda\left(\bigcup_{e\in E^{(i+1,K)}}\mathcal{E}_e^1,\ 1\right)$$

$$\lambda\left(\mathcal{E}_{e_i}^t,\ t\right) = (e_i) + \lambda\left(\bigcup_{e\in E^{(i+1,K)}}\mathcal{E}_e^t,\ t\right)$$

$\square$

**Lemma C.2.** *At round $t$ when $m$ experts have died so far and $e_j \in E_a^t$, $\lambda(\mathcal{E}_{e_j}^t, t)$ is equal to $2^m$ copies of $\lambda(\mathcal{E}_{e_j}^1, 1)$.*

Recall that $\Pi_D$ is the set of all possible orderings of elements in set $D$ of length $|D|$ and similarly, $\mathcal{E}_D$ is the set of all effective orderings with respect to $\Pi_D$.

*Proof.* Due to the constructive building of $\mathcal{E}_{e_i}^1$, $\lambda(\mathcal{E}_{e_i}^1)$ is equal to

$$(e_i) + \lambda\left(\mathcal{E}_{\{e_{i+1},\dots,e_K\}}^1,\ 1\right) = (e_i) + \lambda\left(\bigcup_{i+1\le j\le K}\mathcal{E}_{e_j}^1,\ 1\right) = (e_i) + \bigcup_{i+1\le j\le K}\lambda\left(\mathcal{E}_{e_j}^1,\ 1\right) \quad (18)$$

We use induction on $m$ to prove the claim.

*Induction Basis:* The claim trivially holds when $m = 0$.

*Induction Hypothesis:* When $e_1, e_2, \dots, e_{i-1}$ are dead before round $t-1$, for any $j \ge i$ we have $\lambda(\mathcal{E}_{e_j}^{t-1},\ t-1) = 2^{i-1}\lambda(\mathcal{E}_{e_j}^1,\ 1)$

*Induction Step:* Assume that at round $t-1$, $e_i$ dies.

$$\lambda\left(\mathcal{E}_{e_i}^1,\ 1\right) = (e_i) + \bigcup_{e\in E^{(i+1,K)}}\lambda\left(\mathcal{E}_e^1,\ 1\right)$$

$$2^{i-1}\lambda\left(\mathcal{E}_{e_i}^1,\ 1\right) = (e_i) + 2^{i-1}\bigcup_{e\in E^{(i+1,K)}}\lambda\left(\mathcal{E}_e^1,\ 1\right)$$

$$2^{i-1}\lambda\left(\mathcal{E}_{e_i}^1,\ 1\right) = (e_i) + \bigcup_{e\in E^{(i+1,K)}}2^{i-1}\lambda\left(\mathcal{E}_e^1,\ 1\right)$$

$$\lambda\left(\mathcal{E}_{e_i}^{t-1},\ t-1\right) = (e_i) + \bigcup_{e\in E^{(i+1,K)}}\lambda\left(\mathcal{E}_e^{t-1},\ t-1\right) \quad (19)$$

Where the second and forth equality follows by applying the induction hypothesis to the left and right sides of the first and third line and first equality holds due to Section 3. Therefore when $e_i$ dies, any $\pi \in \mathcal{E}_{e_i}^{t-1}$ we have $\pi \in \mathcal{E}_e^t$ where $e \in E^{(i+1,K)}$ hence

$$\bigcup_{e\in E^{(i+1,K)}}\lambda\left(\mathcal{E}_e^t,\ t\right) = \bigcup_{e\in E^{(i+1,K)}}2\lambda\left(\mathcal{E}_e^{t-1},\ t-1\right) = \bigcup_{e\in E^{(i+1,K)}}2^i\lambda\left(\mathcal{E}_e^1,\ 1\right)$$

where the second equality is from the induction hypothesis. This is easy to see that each set in the union is independent from others, so $\lambda(\mathcal{E}_{e_j}^t,\ t) = 2^i\lambda(\mathcal{E}_{e_j}^1,\ 1)$ where $j > i$. $\square$

### C.3 Proof of Theorem 5.3 (Adapting to the number of nights m)

*Proof of Theorem 5.3.* The idea is to use a simple counting argument. Let $\pi$ be a best permutation expert (it typically will not be unique). For the dying sequence that actually occurs, we will lower bound how many other permutations have the same behavior as this one (we call these behavioral copies). First, observe that if there are $m$ nights, then each permutation expert can only change the actual expert it uses for prediction at most $m$ times (for a total of $m + 1$ different experts). Suppose that, over the course of the game, $\pi$ predicts as $e_{i_1}, e_{i_2}, \dots, e_{i_l}$ where $l \le m + 1$. Now, consider those permutations that actually *begin* as $e_{i_1}, e_{i_2}, \dots, e_{i_l}$. As the first $l$ positions are fixed, there are $(K - l)!$ such permutations and hence at least $(K - l)!$ behavioral copies of $\pi$. Hence, if $\pi$ is the best expert, then we can compete with it using an $\varepsilon$-quantile bound with $\varepsilon = \frac{(K-l)!}{K!} = \prod_{r=0}^{l-1}\frac{1}{K-r} = \varepsilon_l$.

Although we do not know the best choice of $\varepsilon$, as we run Hedge on top of $K$ copies of Hedge-Perm-Unknown and as one of the copies will use learning rate $\varepsilon_l$, we can compete with this optimally tuned copy with additional regret overhead of $\sqrt{T \log K}$. Moreover, a basic quantile bound exercise shows that the regret of the optimally tuned copy will be $\mathcal{O}(\sqrt{T(l+1) \log K})$, where $l \leq m$. $\qquad \square$