[Reviews · NeurIPS 2019]

Reviewer 1



I feel like I have little to say about the paper. It tackles a reasonable problem variation, is well written, and contains some clearly interesting ideas in handling the problem. I enjoyed reading the paper and felt I learned something from it. If there is a possible negative comment it might be that the paper seems arguably "safe" or "incremental" -- it's not going to break open a new area. But it's well written, it covers a natural collection of variations, it has a number of good ideas, and it gets good theoretical results. Good ideas include there being a smaller number of "effective experts", that is not all K! orderings are important. The background is well explained. There is a nicely proven new lower bound. The Hedge-based algorithms are clearly presented as well. The proofs are clearly given in the appendix and are also well presented.

Reviewer 2



The paper investigates an interesting special case of the specialist experts protocol, namely, the case where the experts can sleep from some point on, i.e., die. An algorithm is constructed and upper and lower bounds are provided. The paper appears to have thoroughly studied the question. A particularly strong point of the paper is some non-asymptotic lower bounds for the hedge algorithm. I think this is really interesting work.

Reviewer 3



The dying expert setting is interesting, it would be appreciated to give more examples. The overall writing is good and easy to follow. I have a simple question on the performance measure, ranking regret. In the definition of (1), authors claim \sigma^t(\pi) is the first alive expert of ordering \pi in round t. So why do we need to specify the "first" alive expert, rather than the alive expert with the optimal performance? Meanwhile, a contrary setting -- growing expert -- should be mentioned in the related work. [1] Mourtada, Jaouad, and Odalric-Ambrym Maillard. "Efficient tracking of a growing number of experts." ALT (2017). Both upper bound and lower bound for dying expert setting are established, and this paper distinguishes the hardness of known versus unknown dying order. Minor issues: I would like to mention paper of [2] gives the non-asymptotic lower bound for expert advice problem, though the form is not as neat as Thm 4.1. [2] Orabona, Francesco, and Dávid Pál. "Optimal non-asymptotic lower bound on the minimax regret of learning with expert advice." arXiv preprint arXiv:1511.02176 (2015). line 222: missing one ")"

[Author Response · NeurIPS 2019]

We would like to thank the reviewers for their positive and constructive comments. As mentioned in the paper, the setting of dying experts can be motivated by problems such as fairness in machine learning. Also, interesting follow-up works may include studying the problem in the bandits setting. Below we respond to each of your comments.

**Reviewer 1**

*Comment:* With more space the authors might present more discussion of past/related work

*Response:* Thanks, we will expand our discussion of related work, in particular including references [2]–[4] below.

**Reviewer 2**

*Comment:* It would be interesting to know if the approach of [1] works here and gives similar results.

*Response:* The notion of regret in the "Prediction with specialist experts' advice" section of [1] (this is the relevant section) is *per-action* regret, and so we believe that the results are not directly applicable to our setting (we adopt the notion of *ranking* regret). In per-action regret, the performance of the algorithm is compared to an expert only over those rounds in which that expert was alive. In particular, per-action regret is the difference between the cumulative loss of the algorithm and an expert where the summation is taken over the rounds that expert is available. This makes the notion incomparable to the ranking regret where *all* the rounds are included, and in many settings per-action regret is smaller than ranking regret (see the commentary in [5]). In addition, we would like to mention that there at least can be no direct application of the results of [1] to obtain our results. To see this, let us take Corollary 4 of [1] (which holds for $\eta$-mixable losses) and fix $\eta = \sqrt{\frac{\ln K}{T}}$ to recover a bound comparable to ours. This yields $\mathcal{O}\left(\sqrt{T \ln K}\right)$ which as we mentioned, holds for per-action regret. If a similar bound held for ranking regret, this would contradict the lower bound that we established.

**Reviewer 3**

*Comment:* Why do we need to specify the "first" alive expert, rather than the alive expert with the optimal performance?

*Response:* Thanks for this great question! We are unsure of the right interpretation and so offer three potential interpretations and responses for each. (1) We interpret "the alive expert with the optimal performance" as the best expert that is alive in all the rounds. In this case, the notion of regret becomes much weaker than the one that we use (the ranking regret). (2) We interpret the "optimal performance" as comparing to the best expert in each round (i.e. any expert whose instantaneous loss in that round is the minimum achieved by all the experts in that round). In this case, it is known that achieving sub-linear regret is hopeless. (3) We interpret the "the alive expert with the optimal performance" in each round as the alive expert that is the leader in that round, i.e., the awake expert that has the least cumulative loss from round 1 through to the end of the current round. Looking at the problem this way, one can come up with a two-expert construction (even in the simple case where no one dies) where: the leader alternates between two experts and, in a given round, the leader will always be the expert that obtains the minimum loss in that round; thus, this notion of regret becomes equivalent to the second interpretation and the problem becomes hopeless.

*Comment:* Meanwhile, a contrary setting – growing expert – should be mentioned in the related work.

*Response:* Thanks for the comment. We are aware of this work (also a similar one, [3]) and we will mention them in the related work.

*Comment:* I would like to mention paper of [4] gives the non-asymptotic lower bound for expert advice problem, though the form is not as neat as Thm 4.1.

*Response:* Thanks for pointing out this previous work on the lower bound. We were not aware of this result of Orabona and Pál; while it seems more complicated to prove, we appreciate the fact that their result gives explicit constants. We will be sure to cite this paper.

[1] Chernov and Vovk, "Prediction with expert evaluators' advice". ALT, 2009.

[2] Mourtada, Jaouad, and Odalric-Ambrym Maillard. "Efficient tracking of a growing number of experts." ALT, 2017.

[3] Gofer, Eyal, et al. "Regret minimization for branching experts." COLT, 2013.

[4] Orabona, Francesco, and Dávid Pál. "Optimal non-asymptotic lower bound on the minimax regret of learning with expert advice." *arXiv preprint arXiv:1511.02176* (2015).

[5] Kale, Satyen, Chansoo Lee, and Dávid Pál. "Hardness of online sleeping combinatorial optimization problems." NIPS, 2016.


[Meta-Review · NeurIPS 2019]

A incremental nice work in the experts setting for the case of experts "dying off." In addition to the upper bound the reviewers found the lower bounds of interest. The reviewers are unanimous in their opinion that this paper should be accepted.